# Deep-Learning-Based Annotation Extraction Method for Chinese Scanned Maps

**Xun Rao [1,2], Jiasheng Wang [2,3,*], Wenjing Ran [1,2], Mengzhu Sun [2,3] and Zhe Zhao [1,2]**

1    School of Information Science and Technology, Yunnan Normal University, Kunming 650500, China;
     rx5428@user.ynnu.edu.cn (X.R.); rwenjing@user.ynnu.edu.cn (W.R.); zhaoz@user.yunu.edu.cn (Z.Z.)
2    The Engineering Research Center of GIS Technology in Western China of Education of China, Yunnan
     Normal University, Kunming 650500, China; mzsun@ynnu.edu.cn
3    Faculty of Geography, Yunnan Normal University, Kunming 650500, China
*    Correspondence: jersonwang@ynnu.edu.cn

**Abstract:** One of a map's fundamental elements is its annotations, and extracting these annotations is an important step in enabling machine intelligence to understand scanned map data. Due to the complexity of the characters and lines, extracting annotations from scanned Chinese maps is difficult, and there is currently little research in this area. A deep-learning-based framework for extracting annotations from scanned Chinese maps is presented in the paper. Improved the EAST annotation detection model and CRNN annotation recognition model based on transfer learning make up the two primary parts of this framework. Several sets of the comparative tests for annotation detection and recognition were created in order to assess the efficacy of this method for extracting annotations from scanned Chinese maps. The experimental findings show the following: (i) The suggested annotation detection approach in this study revealed precision, recall, and h-mean values of 0.8990, 0.8389, and 0.8635, respectively. These measures demonstrate improvements over the currently popular models of $-0.0354$ to 0.0907, 0.0131 to 0.2735, and 0.0467 to 0.1919, respectively. (ii) The proposed annotation recognition method in this study revealed precision, recall, and h-mean values of 0.9320, 0.8956, and 0.9134, respectively. These measurements demonstrate improvements over the currently popular models of 0.0294 to 0.1049, 0.0498 to 0.1975, and 0.0402 to 0.1582, respectively.

**Keywords:** scanned map; Chinese recognition; annotation detection; annotation recognition; deep learning

## 1. Introduction

Maps provide visual information about geographic features and locations and are an important visual representation of geographic data. Map labels and other textual details are referred to as map annotations. They are essential components of a map and are vital in assisting users with their geographic queries, thereby helping them find the locations that the map covers and understand its contents. Geographic information science (GIS) technology combines spatial analysis tools and computer science methods with the distinctive visual effects of maps. The functionality of maps is enhanced by this connection. However, at the moment, maps are frequently offered as digital picture files, scanned maps, or paper maps. Information about map annotations cannot be readily extracted by computers from these formats [1]. A key stage in enabling computer intelligence to interpret map information is the extraction of annotations from map pictures. This process is also essential for attaining intelligent map information retrieval.

The extraction of map annotations relied heavily on manual visual interpretation in its early phases. This method was time-consuming, labor-intensive, and inefficient, even if it provided excellent accuracy. This operation would become very difficult and possibly impossible when dealing with a large number of annotations to be taken from maps [1].

Some researchers have suggested a technique where annotations are first identified within maps using computer technology, followed by manual interpretation and conversion, in order to minimize the need for human participation. In the early stages of map annotation localization, techniques such as cluster analysis [2], morphological operations [3], segmentation [4], labeling connected components [2] and the use of image pyramid methods have been used [5]. Even though these techniques are capable of automatically locating map annotations, the extraction of such annotations is less precise when they are intertwined and overlapped with other elements on the map. Li et al. [6] introduced an interactive annotation extraction tool that lets users choose the criteria for character grouping and color separation. The accuracy of extracting map annotations has been somewhat enhanced by this utility. These techniques can automatically position annotations, but the positioning procedure is prone to interference from other map features, which reduces the precision of the map annotation extraction. Additionally, these techniques still rely heavily on manual labor for annotation interpretation and conversion, which leaves the problem of extracting annotations from a large number of maps unresolved.

Some academics have suggested integrating optical character recognition (OCR) technology to address the problem of automatic interpretation and the conversion of map annotations. OCR technology can examine and identify text-based image files, thereby extracting the text and layout information. In order to do this, the text must be identified inside the image and returned in textual form. Pouderoux et al. [7] and Pezeshk et al. [1] used OCR software to identify discovered map notes. However, due to the limits of conventional OCR recognition techniques, input images must be at least 300 dpi, and characters must be sufficiently large for conventional OCR technology to recognize them [7]. Furthermore, the problem of problematic annotation localization when annotations overlap and tangle with other map characteristics is still unaddressed by current methods. Chiang et al. [8] suggested using a manual user acquisition of map annotations to obtain user labels as part of a supervised learning technique. The characteristics of the annotations are learned by making use of the acquired user labels. Although it might not be appropriate for supporting jobs requiring the extraction of a sizable number of annotations, this method improves annotation extraction accuracy through user involvement.

The completion of OCR tasks has advanced significantly with the advent and development of deep learning, as well as through the effective combination of OCR technology and deep learning [9,10]. It was proposed by Li et al. [11] to use the FRCNN model for annotation localization. Through graph segmentation and clustering processes, annotations were subsequently isolated from other map features. In order to automate the extraction of map annotations, the Google Tesseract OCR Engine was then used to recognize the annotations. The recognition was then improved using a geographic database. Additionally, Zhai et al. [12] effectively constructed a model and dataset integrated transfer learning strategy to identify unstructured map text.

In conclusion, the available study raises the following problems: (1) The majority of recent research has been devoted to extracting annotations from English maps. Chinese characters are numerous and structurally difficult compared to English characters, which has hindered research on Chinese annotation extraction. (2) When compared to other types of text extraction (such as text from handwritten notes, documents, scenes, etc.), map annotation extraction is distinguished by its high quantity, nonuniform distribution, and susceptibility to influence from map line features. (3) There are not enough publicly accessible datasets of Chinese maps that can be used to train deep learning models at the moment.

The paper offers a deep-learning-based method for the automatic extraction of annotations from scanned Chinese maps in order to overcome the aforementioned problems. For detecting annotations in scanned Chinese maps and identifying them in scanned Chinese maps, this method principally uses an improved EAST model and a CRNN model based on transfer learning. Among them, annotation detection is used to locate the position of the annotation in the image and return the general extent of the annotated area; annotation recognition refers to the conversion of the annotated image in the scanned map

into text, i.e., the text in the image is converted from the form of a picture into the form of a computer-readable text.

## 2. Materials and Methods

### 2.1. Dataset Construction

Chinese scanned map databases were not readily available; therefore, this study's dataset had to be made. Real scanned map datasets and simulated map datasets make up the majority of the dataset. While the simulated map dataset was largely used for model training, the real scanned map data was mostly used for model validation and evaluation.

### 2.1.1. Real Scanned Map Dataset Construction

The Chinese maps used in this study are from the "Atlas of Yunnan Province 2002 Edition". From this source, a total of 45 maps were chosen, and they were manually scanned to produce the Chinese scanned map images. A total of 45 different Chinese scanned map images were created after manually filtering and cropping the acquired Chinese scanned map photos, which contained some useless information. The resulting Chinese scanned map images were then manually annotated using the labeling program. After completing these stages, the final dataset of actual scanned map images was obtained and mainly used for the model's evaluation and validation.

### 2.1.2. Simulated Scanned Map Dataset Construction

It is challenging to obtain simulation maps with the same style as all of the genuine scanned maps in this work, since the real scanned maps in this study comprise a variety of map styles. Given that the majority of the actual scanned maps in this study are from the standard cartography, which adheres to a set of specifications, and that their annotation styles are similar, this paper complicated the other elements while creating simulation maps to make the trained models as adaptable to as many various map styles as possible. Figure 1 shows a comparison of the annotation styles of some of the real scanned maps and the simulated maps.

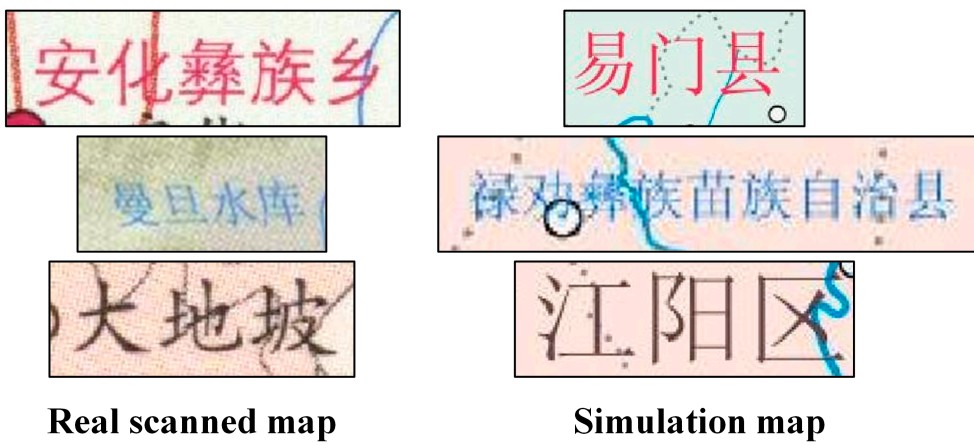

**Real scanned map**          **Simulation map**

**Figure 1.** Comparison of real scanned map and simulated map styles.

The paper developed a simulated map generator to produce simulated map datasets in batches in accordance with the properties of the maps. Figure 2 depicts the workflow of the generator, which can be distilled into four key steps:

(1) Map base selection: For this study, three different types of map bases were selected: web photos, maps of urban street scenes, and maps of administrative areas. The latter was picked to improve the dataset, whereas the first two were mostly selected to imitate popular map types. Annotations can be affected by other map components (particularly line features) during the extraction process. The robustness of the learned model was increased by including more complicated backdrops during training.



(2)     Annotation addition: Specific properties (content, font, text color, and size) were used to introduce annotations. Annotations on standard maps typically fall under a number of predefined categories that are present on the same map, though these categories may vary from one map to another. In particular, rather than employing a few predetermined categories of characteristics during the development of training set maps, annotation attributes were produced randomly to improve model generalization. The placement of the annotations on the maps was random, and a straightforward overlap detection algorithm was developed to avoid overlap and covering between generated annotations:

  a.     Create an image of the annotation region that is the same size as the target map in order to capture the locations of the created annotations.
  b.     Create predefined annotation areas after obtaining annotation attributes.
  c.     Evaluate the annotation area image against the predefined areas until there is no intersection between the predefined area and the annotation area image; acquire a new predefined area if they do.
  d.     In the annotation area image, note the predefined area.

(3)     Image Augmentation: Images created by computers frequently have superior quality and less noise interference. However, when it comes to scanned maps, the scanning procedure can be impacted by human or mechanical mistakes, with measurable effects on the quality of the scanned map. Several problems that are frequently encountered during scanning were identified and introduced as disturbances to the simulated maps in order to more accurately reflect the conditions of real scanned map images:

  a.     Blurriness is one issue that can arise from low scanning resolution, which makes it challenging to show details accurately.
  b.     Color shifts: Scanned images may show color shifts, in which the image's colors differ from those of the actual object. This is frequently caused by problems with the scanner's or scanning software's color calibration. Another issue that could arise during scanning is shadows or reflections.
  c.     Noise: Patterns might become distorted and grainy due to random noise brought on by elements such as the scanner's light source, optical path, and sensor.

  Different forms of disturbances were randomly blended and added to the simulated maps after annotations had been added.

(4)     Simulated map and label generation: Both the content and spatial details of the annotations that were created in step (2) were recorded. To develop labels for annotation detection and identification, these were later combined. In parallel, a new picture was created by copying all annotation characteristics to it, with the exception of text color. The annotation separation label was made using this newly constructed image. The final simulated maps were obtained once image augmentation step (3) was finished. These final simulated maps were linked to the generated labels. In the end, the output included the simulated maps and the labels that went with them.

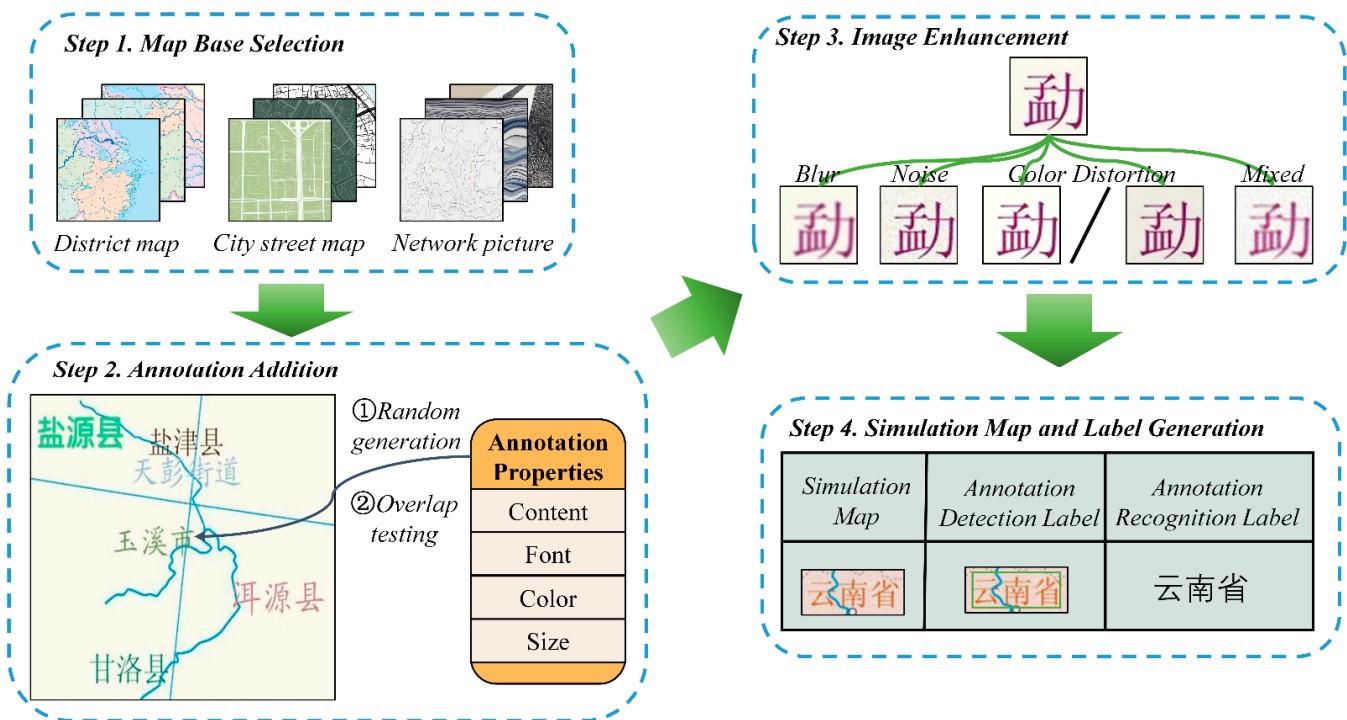

**Figure 2.** Flowchart of the simulation map generator.

### 2.2. Scanned Map Annotation Extraction Framework

This paper accomplished the extraction of Chinese scanned map annotations through three steps, as illustrated in Figure 3. The process involves three steps. In the first step, data preprocessing is carried out, which involves cropping or resizing the scanned map to meet the requirements of the annotation detection model. This is because, in the process of scanning, there will be a lot of unavoidable factors that lead to poor image quality, and each scanning according to the impact is not the same, which produces a wide range of image noise categories, thereby leading to the fact that it is difficult to ensure that the scanning of the map quality is consistent; therefore, this method is not set to denoise the process, but rather the impact of the noise is added to the model training, such as was performed in this paper in the generation of simulated maps when the interference of the noise was added. In the second step, annotation detection is performed using a trained model, which is applied to the preprocessed scanned map to detect annotations. This yields the coordinates of the annotations (if cropping was performed during preprocessing, it is recommended to adjust the coordinate offsets based on cropping rules). In the third step, annotation recognition is conducted. A trained annotation recognition model is applied to the results of the annotation detection step to recognize the annotations, thereby producing corresponding recognition outcomes. By combining these results with the annotation detection outcomes, the final extracted annotation results are ultimately generated.

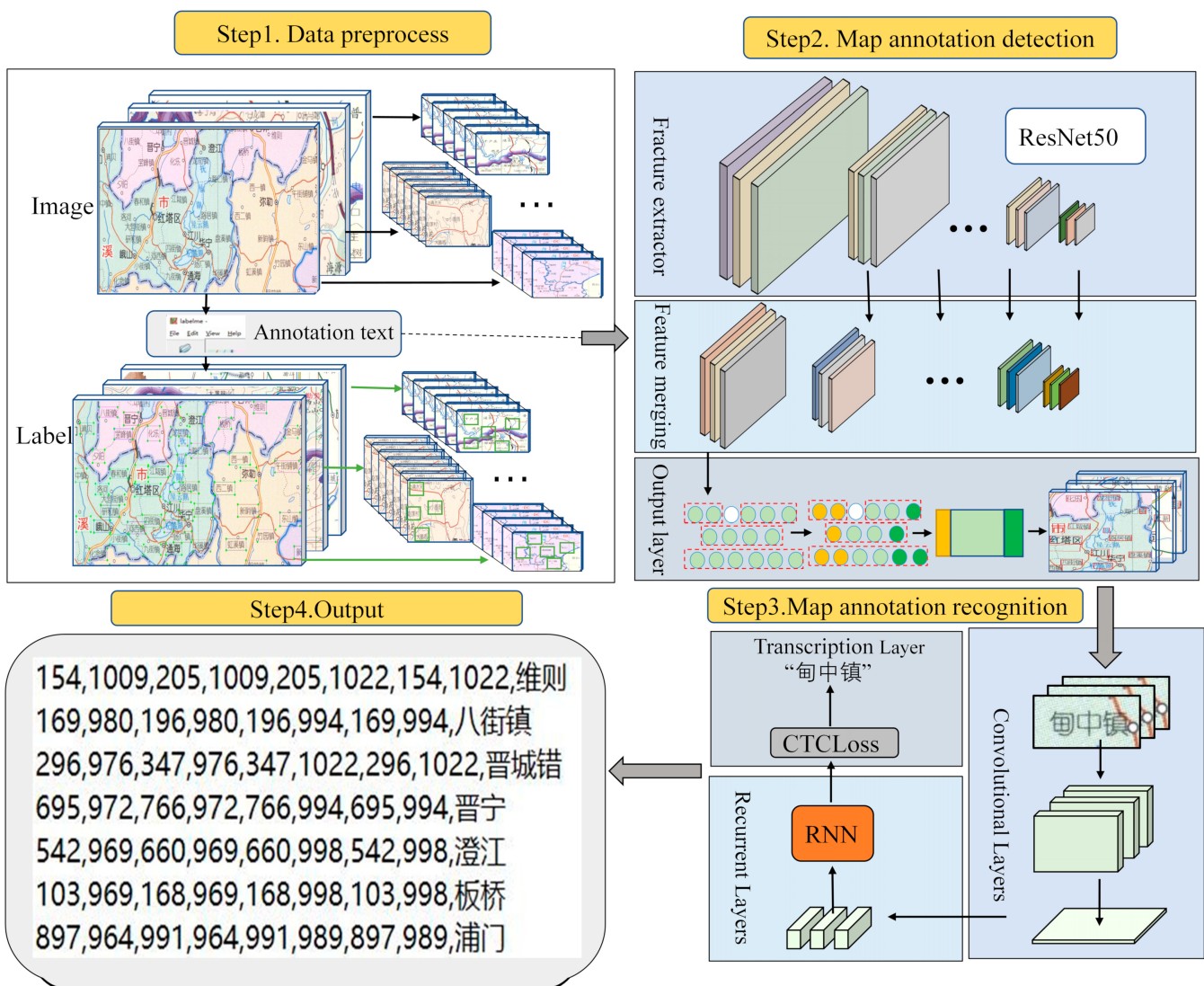

**Figure 3.** Methodology framework diagram.

### 2.3. Improved EAST Annotation Detection Model

The EAST [13] model's structure was adopted by the annotation detection model in this paper. Three modules—feature extraction, feature fusion, and output—make up the overall model, as shown in Figure 4. The EAST model's basis is the foundation upon which the main alterations in this study have been made. A ResNet was used to extract deep image features in the feature extraction module, thereby allowing for the acquisition of profound picture characteristics. An ASF (adaptive spatial fusion) module was added to the feature fusion module to fuse features from different levels and create multiscale annotation features. Additionally, the output module incorporated the AdvancedEAST structure to improve the detection effectiveness for lengthier texts.

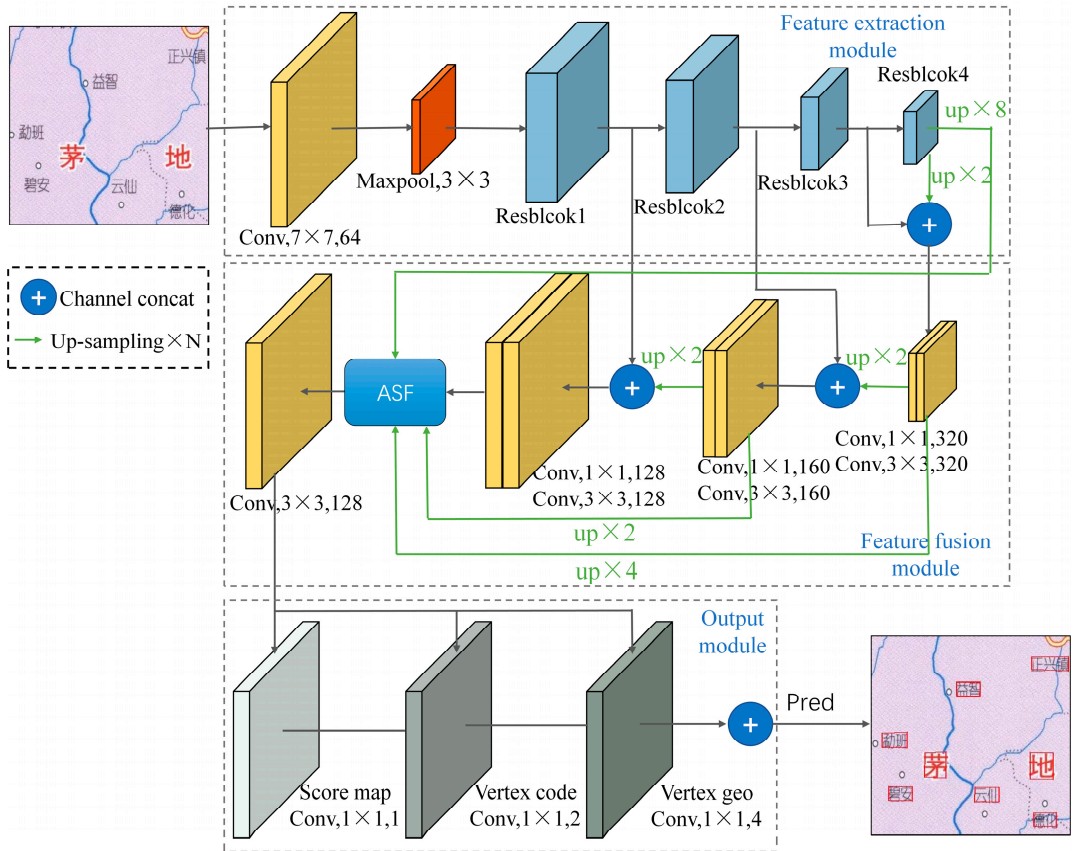

**Figure 4.** Improved AdvancedEAST notation detection model structure.

(i) Feature Extraction Module

ResNet50 [14] was used in this study as the feature extraction network. Through the utilization of residual structures, ResNet reduces the problems of disappearing gradients and network deterioration brought on by excessively deep networks. Deeper networks can be used to extract features, thereby allowing for the capture of features at several levels. The retrieved features grow increasingly abstract and semantically rich as the network depth rises.

The ResNet50 average pooling layer and fully connected layer, which are employed for classification tasks, were not included in the model. Instead, the feature fusion section used the outputs of the four residual modules within ResNet50 as its inputs. Figure 5 shows the generated residual modules.

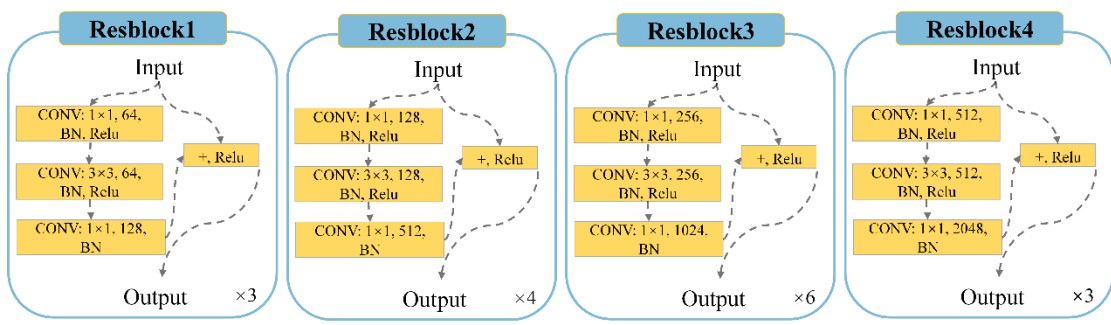

**Figure 5.** Resblock structure diagram.

(ii)   Feature Fusion Module

The main characteristics of map annotations are their designs and styles. These traits are mostly represented in channel features and spatial features in image processing. As a result, the feature fusion module in this article included an ASF module [15]. With this feature, the model's annotation detection accuracy was aimed to be improved.

Figure 6 depicts the layout of the adaptive spatial fusion (ASF) module. It should be noted that only the spatial attention module was used in the [15] study. However, this work also included a channel attention module because of the unique qualities of map annotations. The findings of Ref. [15] from 2022 are consistent with the underlying ideas. The channel spatial attention module is where the difference is found. In the channel spatial attention module, the intermediate feature $S$ undergoes channelwise feature computation, thus yielding the channel feature $A_C \in R^{N \times C}$; this feature is elementwise multiplied with $S$, thereby resulting in an intermediate feature $S'$ that incorporates channel weights. Subsequently, spatial feature computation is performed on $S'$, thereby producing the spatial feature $A \in R^{N \times H \times W}$. Finally, $A$ is elementwise multiplied with $S$, thereby generating the weighted feature map $F \in R^{N \times C \times H \times W}$. Here, the definitions of the features are as follows:

$$
\begin{aligned}
S &= Conv(concat([X_0, X_1, \ldots, X_{N-1}])) \\
A_C &= Channel\_Attention(S) \\
S' &= S \oplus A \\
A &= Spatial\_Attention(S') \\
F &= concat([E_0 X_0, E_1 X_1, \ldots, E_{N-1} X_{N-1}])
\end{aligned}
\tag{1}
$$

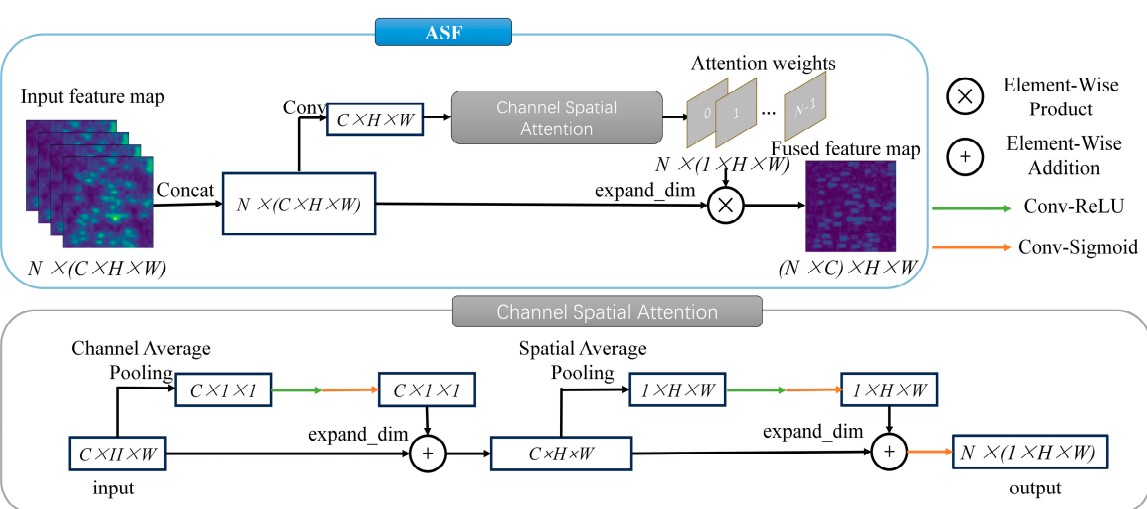

**Figure 6.** ASF structure diagram.

(iii)   Output Module

The method used in this study incorporated the output module from AdvancedEAST (https://github.com/huoyijie/AdvancedEAST accessed on 1 July 2023), wherein it took into account that longer texts are frequently found inside maps and that the EAST model's performance with respect to identifying long texts may not be as good as it could be. This was done to improve the model's ability to detect and handle lengthy textual comments in map pictures.

AdvancedEAST is an enhanced text detection algorithm built upon the foundation of the EAST model. It addresses the limitations of EAST in detecting long texts. By leveraging the EAST network architecture, AdvancedEAST cleverly designs a loss function based on text bounding boxes. This transformation shifts the challenge of long text detection into the task of detecting the boundaries of the text's head and tail.

In contrast to EAST, AdvancedEAST's output structure is made up of three primary parts: the score map, vertex code, and vertex geometry. Figure 4 provides an illustration of this architecture. The score map represents the confidence level, thereby indicating the probability of a point being within the text box. The vertex code consists of two parameters. The first parameter signifies the confidence level that a point is a boundary element. The second parameter indicates whether the point belongs to the head or tail of a text element. Vertex geometry provides the coordinates of two predictable vertices for boundary pixels. All pixels together constitute the shape of the text box. Regression vertex coordinates are predicted only for boundary pixels. Boundary pixels encompass all pixels belonging to the head and tail. The prediction of the vertex coordinates for the short sides of the head or tail is achieved through the weighted average of the predicted values of all boundary pixels. Two vertices each are predicted for the boundary pixels of the head and tail, thereby resulting in a total of four vertex coordinates. Since the input image dimensions may not match the dimensions of the original image, the XY coordinates obtained here are not actual coordinates. Instead, they represent the offset of the current point's XY coordinates.

### 2.4. CRNN Annotation Recognition Model Based on Transfer Learning

Chinese scanned map recognition presents two distinct difficulties when compared to typical document text recognition tasks: Chinese scanned maps have more complex background interference and line element interference, especially with line elements, than English letters do. Chinese characters, in contrast to English letters, are more complex and numerous, with over 2500 commonly used characters and over 1000 less frequently used characters. It can be challenging to tell these line element attributes apart from annotation features because they frequently resemble one another. Chinese characters have a lot of visual similarities, unlike English letters; therefore, even a small bit of interference can rapidly result in skewed recognition results.

CRNN [16] is a deep learning model for processing sequence data, especially for text recognition and OCR tasks. CRNN combines the strengths of the convolutional neural network (CNN) and recurrent neural network (RNN) to make it excellent at processing variable length sequence data (such as lines of text or paragraphs).

Transfer learning is a machine learning method that assists algorithms in acquiring new knowledge by leveraging the similarity between prior knowledge and new knowledge. Transfer learning algorithms can be classified into four categories [17]: Instance-based transfer is the first category—this approach aims to build a reliable learning model by selecting instances from a source domain; feature-based transfer is the second category—in this category, attempts are made to discover shared feature representations between the source and target domains; parameter-based transfer is the third category—this involves identifying common parameters or prior distributions between source and target data to achieve knowledge transfer; and elation-based transfer is the fourth category—this type mainly deals with nonindependent and identically distributed data that have existing relationships.

The method used in this paper is parameter-based transfer learning. A dataset of 3.6 million Chinese characters can be found in the source domain at (https://github.com/senlinuc/caffe_ocr accessed on 1 July 2023). To create a pretrained model, the method begins with pretraining on the source domain. Then, the model of the target domain is given the parameters from the first n layers of this pretrained model. The subsequent layers are then adjusted using information from the target domain. The model's performance on the target domain is improved, and the discrepancies between source and target domain data are decreased as a result of the fine-tuning procedure. Figure 7 depicts the transfer learning process.

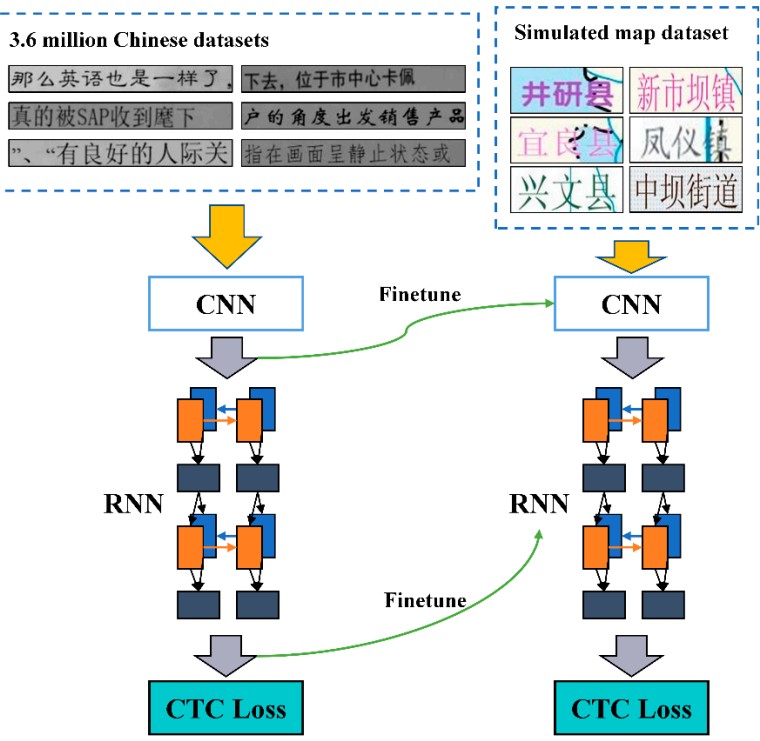

**Figure 7.** Transfer learning process diagram.

*2.5. Evaluation Metrics*

In order to assess the performance of the model, this paper employs three evaluation metrics: precision, recall, and h mean. Their definitions are as follows:

$$\text{precision} = \frac{\text{TP}}{\text{TP} + \text{FP}} \tag{2}$$

$$\text{recall} = \frac{\text{TP}}{\text{TP} + \text{FN}} \tag{3}$$

$$\text{h mean} = \frac{2 \times \text{precision} \times \text{recall}}{\text{precision} + \text{recall}} \tag{4}$$

Among these, TP represents the count of correctly predicted positive samples, FP represents the count of incorrectly predicted positive samples, and FN represents the count of incorrectly predicted negative samples. In annotation detection evaluation, when the predicted region aligns with the actual region, it is considered to be a correct prediction. Specifically, due to the typically large dimensions of complete scanned maps, annotation detection models cannot predict the entire image at once. Instead, they require a sliding window approach for prediction. This unavoidably results in some complete annotations being divided into multiple parts, thus causing the number of predicted annotations to be greater than or equal to the number of annotations in the label data. As a result, the accuracy of annotation identification would not be accurately reflected if conventional text recognition criteria were used (where the entire label must be predicted properly to be considered valid). Additionally, given that the majority of annotations lack strong semantic information, even if they are divided into numerous separate characters, as long as their spatial placements on the map are identical, this would not have an impact on how users read and comprehend the annotations. As a result, this paper proposes the criterion that successfully predicting independent characters is deemed as a correct prediction for the evaluation of annotation recognition.

## 3. Experimental Design

### 3.1. Experimental Environment

The environment used in this experiment was the following: the CPU of the server was a AMD Ryzen 7 5800X @ 3.80 GHZ, the GPU was a NVIDA GeForce RTX3090, the operating system was Windows 10, and the compilation environment was Python 3.8.10, Pytorch 1.9.0, and CUDA 11.2.

### 3.2. Experimental Design

(i) Map annotation detection experimental design: To validate the effectiveness of the proposed model, this paper conducted experiments with a total of eight models, including the one designed in this study and six baseline models. The baseline models were EAST [13], PSE [18], FCE [19], SAST [20], DBNet [21], DBNet++ [15], and TCM-DBNet [22]. All of these models were trained using the same training dataset. In particular, it is important to note that the simulated maps were used as a training set only, while the real maps were used as a training set only. In this paper, the real scanned maps were divided into three groups according to the different levels of complexity of their backgrounds: the first group is the simple line group, whose backgrounds consist mainly of large solid color areas and a small number of lines; the second group is the moderately complex line group, whose backgrounds contain a certain amount of lines; and the third group is the complex line group, whose backgrounds consist mainly of a large number of lines made up of topographic maps similar to topographic maps, and where a large number of lines appear to be interspersed with the annotations. An example of the grouping is shown in Figure 8.

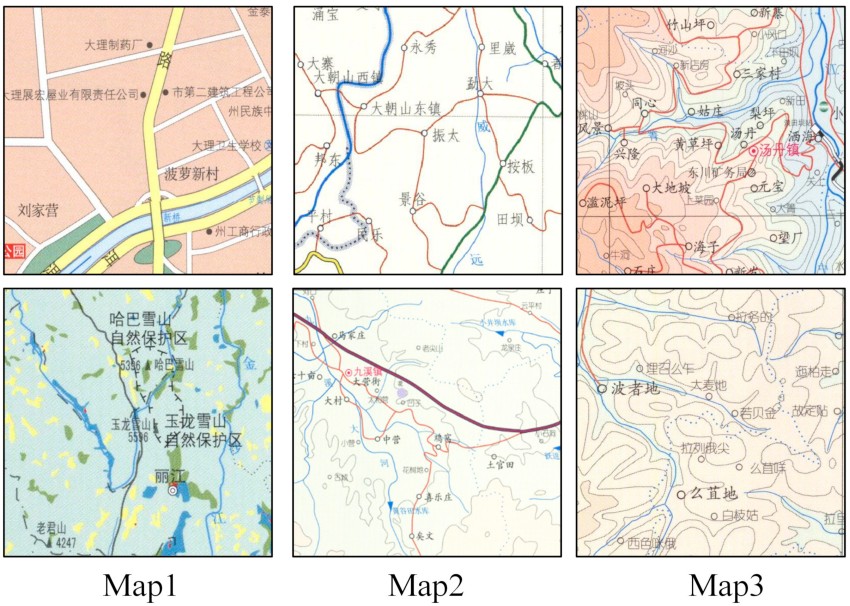

<div align="center">Map1            Map2            Map3</div>

**Figure 8.** Annotation detection results for different annotation styles.

(ii) Map annotation recognition experiment design: To validate the effectiveness of the model, this paper established four control groups in addition to the model designed in this paper. These control groups included the CRNN [16], without transfer learning, trained on a dataset of 3.6 million Chinese characters, as well as three Chinese text recognition OCR projects. The ddddOCR (https://github.com/sml2h3/ddddocr accessed on 1 July 2023) is a recognition library for captchas, the EasyOCR (https://github.com/JaidedAI/EasyOCR accessed on 1 July 2023) is an OCR project that supports multi-language recognition, and the CnOCR (https://github.com/breezedeus/CnOCR accessed on 1 July 2023) is a project specifically designed for Chinese text recognition.

## 4. Results and Discussion

Because the size of the whole scanned map is much larger than the input to the model, if the whole image is directly input to the model, it does not obtain the expected results. Therefore, it is necessary to split the whole scanned map into several parts for prediction and then merge the prediction results to get the final prediction results. In addition, in order to prevent certain annotations from being split when cropping, which affects the prediction results, this paper chose to use two sliding windows for cropping, which have different starting points and the same step size and dimensions; it predicted the results obtained from both of the sliding windows and combined the results obtained from the two sliding windows to obtain the final prediction results.

### 4.1. Experimental Results of Chinese Scanned Map Annotation Detection

Table 1 displays the final annotation detection findings. It is clear from Table 1 that the upgraded EAST model suggested in this paper performed significantly better than the original EAST model. With improvements of 0.4 or greater, the improvement was significant across all metrics. Among the comparable models, the model suggested in this research received the highest overall results in terms of the recall and h mean. Additionally, it had excellent precise performance.

**Table 1.** Comparison of the results of different annotation detection methods. The bolded portion indicates the optimal value in the comparison term.

| Method | | EAST | PSE | FCE | SAST | DBNet | DBNet++ | TCM-DBNet | OWN |
|---|---|---|---|---|---|---|---|---|---|
| | precision | 0.5950 | 0.8061 | **0.9301** | 0.8429 | 0.8682 | 0.9285 | 0.8805 | 0.8984 |
| Map Group 1 | recall | 0.6824 | **0.8494** | 0.8029 | 0.6290 | 0.7246 | 0.7909 | 0.7544 | 0.7633 |
| | h mean | 0.6357 | 0.8272 | **0.8618** | 0.7204 | 0.7899 | 0.8542 | 0.8126 | 0.8209 |
| | precision | 0.4521 | 0.8484 | 0.8876 | 0.8793 | 0.8954 | **0.9534** | 0.9154 | 0.9058 |
| Map Group 2 | recall | 0.5326 | 0.8304 | 0.7330 | 0.5874 | 0.5679 | 0.6429 | 0.6011 | **0.8805** |
| | h mean | 0.4891 | 0.8393 | 0.8029 | 0.7043 | 0.6950 | 0.7680 | 0.7257 | **0.8910** |
| | precision | 0.3886 | 0.7861 | 0.9182 | 0.7890 | 0.8371 | **0.9274** | 0.8435 | 0.8953 |
| Map Group 3 | recall | 0.4408 | 0.8061 | 0.6808 | 0.5065 | 0.5289 | 0.5247 | 0.5128 | **0.8684** |
| | h mean | 0.4130 | 0.7960 | 0.7819 | 0.6170 | 0.6483 | 0.6702 | 0.6378 | **0.8777** |
| | precision | 0.4691 | 0.8083 | 0.9140 | 0.8290 | 0.8618 | **0.9344** | 0.8735 | 0.8990 |
| All | recall | 0.5395 | 0.8258 | 0.7322 | 0.5654 | 0.5997 | 0.6378 | 0.6106 | **0.8389** |
| | h mean | 0.5018 | 0.8168 | 0.8121 | 0.6716 | 0.7043 | 0.7525 | 0.7147 | **0.8635** |

Additionally, text detection models are typically divided into segmentation- and regression-based text detection models. Both EAST and the suggested model fit into the category of regression-based annotation identification models in the experiments carried out in this research. On the other side, segmentation-based annotation detection models include the PSE, FCE, SAST, DBNet, DBNet++ and TCM-DBNet. It is clear that segmentation-based annotation detection models, with the exception of PSE, typically have higher precision but lower recall, which results in lower total h mean scores. This paper makes an educated guess as to the reasons for the performance disparities between most segmentation-based annotation detection models and regression-based models by examining the model structures. Predictions are frequently pixel-based in segmentation-based models, which in certain cases results in more accurate final predicted regions. This strategy could, however, result in the loss of some macroscopic data. Given that lines and annotations at the local level resemble each other quite a bit, the models may unintentionally forecast certain lines as annotations, which would lower the recall value. Regression-based models, on the other hand, frequently use nonmaximum suppression (NMS) algorithms to combine and filter several prediction boxes, thereby resulting in a final prediction box. The precision of this method may not be as high as that of segmentation-based models, but it is less prone to interference from lines, thus producing a more stable recall value. Despite its overall effectiveness, the EAST had a recall value that was substantially higher than its precision value. Although the PSE is a segmentation-based detection model, no other segmentation-based

detection model had a lower recall. It is hypothesized that this is because the multiscale prediction method, which is similar to NMS, was utilized in the prediction stage, which accounted for the precision and recall values being reasonably steady and near.

The results of several annotation detection models applied to diverse annotation styles are shown in Figure 9. Overall, the suggested annotation detection model's detection findings in this research were the most similar to the labels found in the real world. Different styles of map annotations could be successfully detected. While the other annotation identification algorithms are capable of recognizing annotations with a variety of styles, they performed poorly when it came to detecting map annotations with blue text, which frequently designates rivers or lakes.

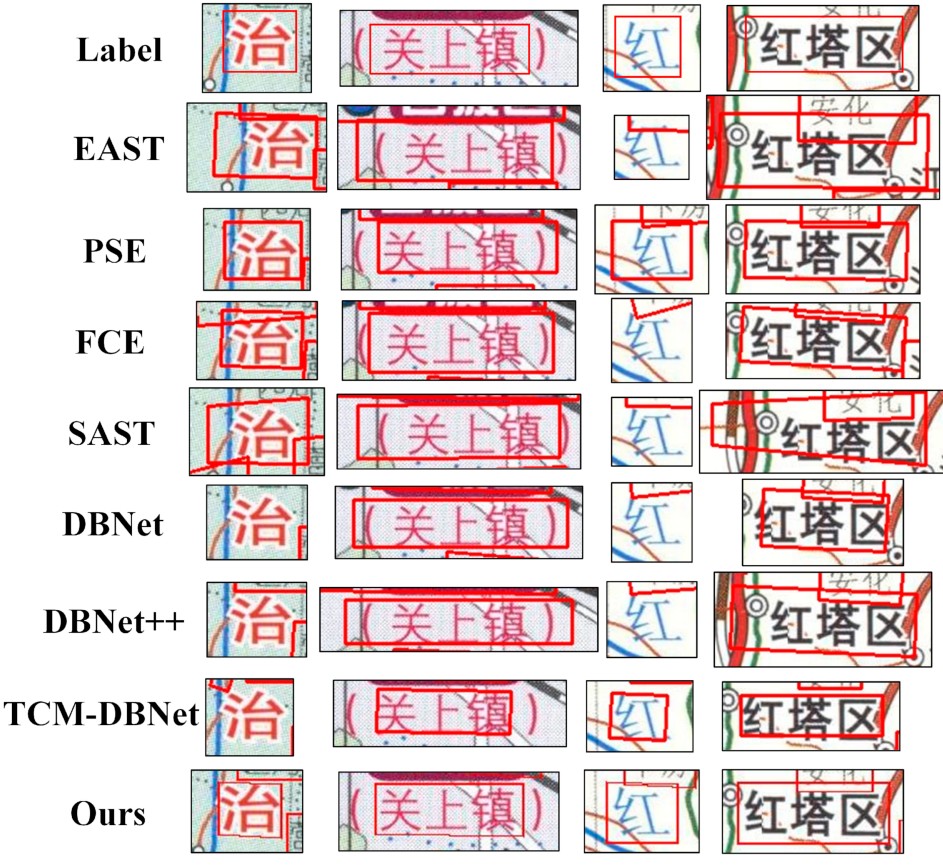

**Figure 9.** Annotation detection results for different annotation styles.

The results of several annotation detection models under varied background interferences are shown in Figure 10. It can be seen that many annotation detection techniques struggled to accurately detect map annotations when the map background was complicated, especially when the annotations overlapped and tangled with the line characteristics. The proposed annotation detection model in this paper, however, successfully resolved this problem. Even in situations where the interference from the map background was severe, it could reliably recognize map annotations.

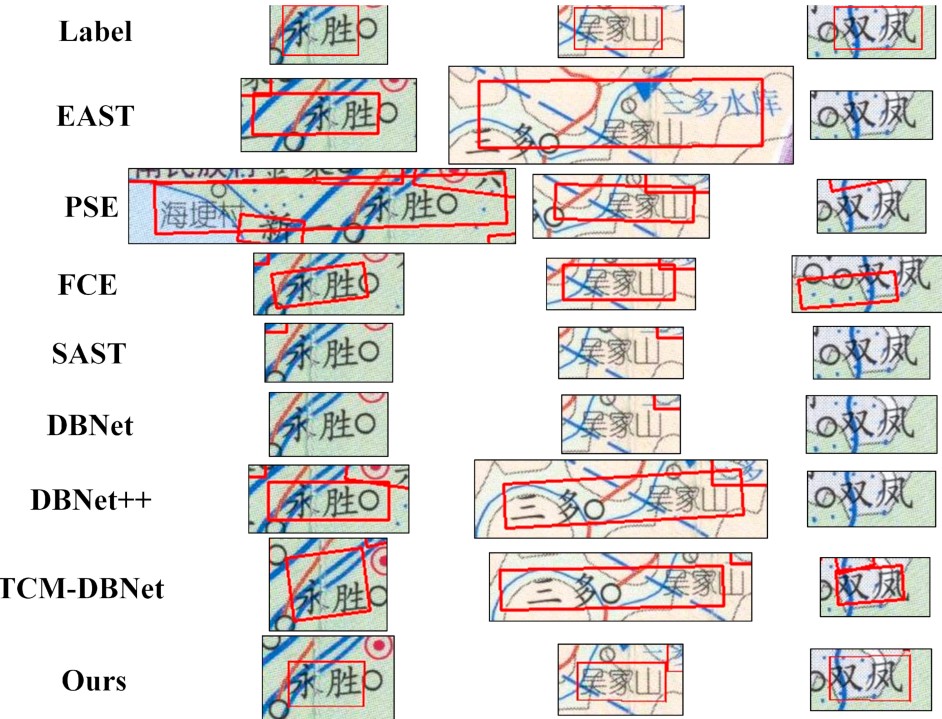

**Figure 10.** Annotation detection results with different map background interference.

*4.2. Experimental Results of Chinese Scanned Map Annotation Recognition*

Table 2 displays the final annotation recognition outcomes. The performance of the CRNN model with transfer learning was better than that of the CRNN model without transfer learning, as can be seen in Table 2. There could be two primary causes for this enhancement: (1) Interference from the lines: One of the main factors impacting its performance is the interference brought on by the lines. While annotation images and document text annotation images are similar, annotation photos frequently have more interference from lines. Due to the similarities between lines and annotations, the CRNN model's lack of transfer learning results in subpar annotation recognition accuracy. (2) Limited character dictionary: Because there are so many Chinese characters, even training on a dataset of 3.6 million Chinese characters might not completely cover all of the characters that could be used. The map dataset chosen for this study contains a number of uncommon Chinese characters that are not frequently used in spoken Chinese. As a result, the CRNN model without transfer learning finds it challenging to infer the appropriate Chinese characters.

**Table 2.** Comparison of the results of different annotation recognition methods. The bolded portion indicates the optimal value in the comparison term.

| Method | | CRNN | ddddOCR | EasyOCR | CnOCR | Ours |
|---|---|---|---|---|---|---|
| | precision | 0.6402 | 0.9066 | 0.8645 | 0.8345 | **0.9336** |
| Map Group 1 | recall | 0.6285 | 0.8604 | 0.8378 | 0.7675 | **0.9065** |
| | h mean | 0.6343 | 0.8829 | 0.8509 | 0.7996 | **0.9199** |
| | precision | 0.6323 | 0.8966 | 0.7545 | 0.8634 | **0.9287** |
| Map Group 2 | recall | 0.6124 | 0.8532 | 0.5745 | 0.7445 | **0.8846** |
| | h mean | 0.6222 | 0.8744 | 0.6523 | 0.7996 | **0.9061** |
| | precision | 0.6148 | 0.9033 | 0.8434 | 0.8443 | **0.9329** |
| Map Group 3 | recall | 0.6212 | 0.8310 | 0.6713 | 0.7573 | **0.8943** |
| | h mean | 0.6180 | 0.8656 | 0.7476 | 0.7984 | **0.9132** |
| | precision | 0.6272 | 0.9026 | 0.8271 | 0.8462 | **0.9320** |
| All | recall | 0.6212 | 0.8458 | 0.6981 | 0.7572 | **0.8956** |
| | h mean | 0.6241 | 0.8732 | 0.7552 | 0.7991 | **0.9134** |

Among all of the control groups, the annotation recognition model utilized in this study performed the best in terms of the various assessment criteria, both at the local and global levels. Additionally, the ddddOCR outperformed the other two OCR projects by more than 0.12 points across all evaluation metrics. The fact that the CnOCR and Easy-OCR are both general-purpose OCR systems with a focus on scene text and document text recognition is probably the cause of this success. However, the ddddOCR concentrates mostly on CAPTCHA recognition. Images used for CAPTCHAs and annotations are similar in that they both frequently have a lot of lines and background distractions. Because CAPTCHAs and annotation images are comparable, this is likely one of the reasons why the ddddOCR's recognition performance outperformed that of the other two OCR systems and closely resembled the model employed in this study.

The recognition outcomes of several annotation recognition algorithms under varied background interference situations are principally shown in Figure 11. In the first image, the character '镇' partially overlapped with map line features, thereby leading to incorrect recognition by some of the recognition models. In the second image, the character '楞' overlapped with two different types of map line features, with significant overlap on its left side. In this case, some recognition models exhibited varying degrees of misrecognition. In the third image, despite the presence of only a few lines as interference, the left-side line caused some recognition models to mistake the character '老' for '佬'. In the fourth image, the character '广' was extensively covered by line features, thus causing some recognition models to fail to recognize it as a character. However, the model put forward in this paper was able to correctly identify annotation data in the aforementioned situations.

| Label | CRNN | ddddOCR | CnOCR | EasyOCR | Ours |
|---|---|---|---|---|---|
| 晋城镇 | 晋诚挂具 | 晋城镇 | 晋城镇R | 晋城=) | 晋城镇 |
| 楞口 | 烤口 | 楞口 | 树口 | __ | 楞口 |
| 老爷山 | 老爷山 | 老爷山 | 佬爷山 | 佬爷_ | 老爷山 |
| 广济 | _济 | _济 | 文济 | __ | 广济 |

**Figure 11.** Annotation recognition results with different background interference.

### 4.3. Chinese Scanned Map Annotation Extraction Error Analysis

The ultimate approach for extracting Chinese scanned map annotations can be accomplished by integrating the models that were obtained in accordance with Figure 3. The whole results of the Chinese scanned map annotation detection are shown in Figure 12, while the partial results of the Chinese scanned map annotation recognition are shown in Figure 12, where the red box indicates a correct detection, the blue box indicates an incorrect detection, and the green box indicates a missed detection.

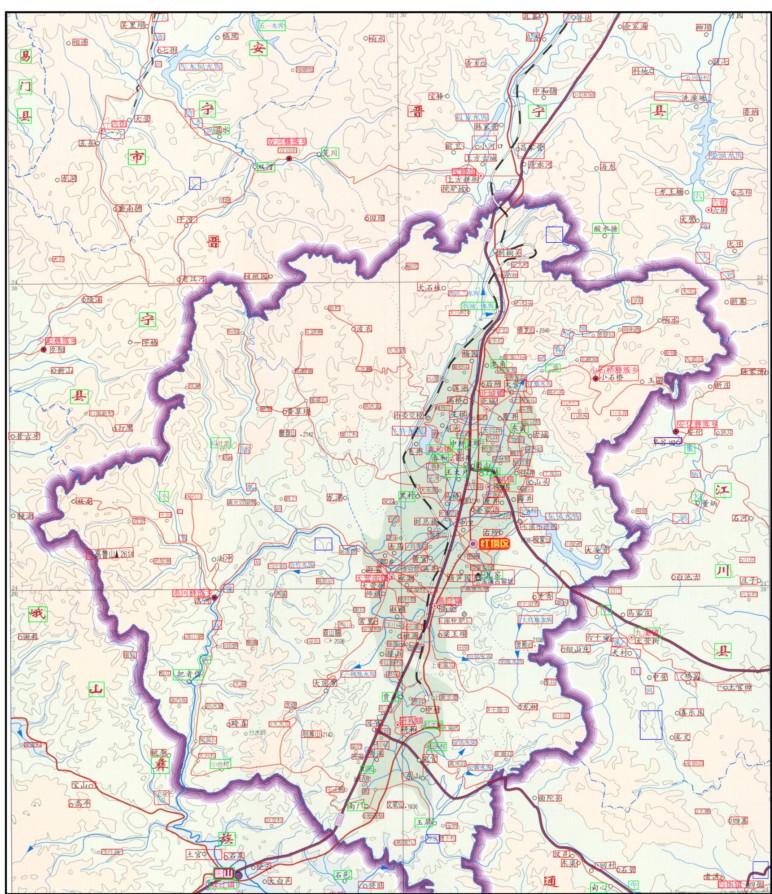

**Figure 12.** Chinese scanned map annotation detection results. The red box indicates a correct detection, the blue box indicates an incorrect detection, and the green box indicates a missed detection.

From Figure 12, it can be observed that the model proposed in this paper was capable of accurately detecting the positions of most of the annotations. However, the annotation detection results for larger annotations were not as satisfactory, as seen in annotations such as '江', '川', and '县' on the far right of Figure 12. This could potentially be attributed to the significant differences between these types of annotations and others. Insufficient samples of this particular annotation style in the training dataset might have contributed to this issue.

In addition, in the middle of Figure 12, we can see that there is a large number of annotations gathered, which was also a region of a high frequency of leakage detection. After analyzing, there are two main reasons for this situation: firstly, because the simulation maps generated in this paper almost did not have this kind of large number of concentrated annotations, this made the training for this case incomplete; secondly, because of the characteristics of the NMS algorithm, some annotations were too close together, which led to the removal of lower scoring prediction frames in the filtering of the prediction boxes so that the lower scoring prediction boxes were removed.

Figure 13 shows the annotation detection results of different annotation detection models, with red boxes indicating correct detection, blue boxes indicating wrong detection, and green boxes indicating missed detection. From the figure, it can be seen that the EAST had a large number of misdetections and missed detections, and the PSE, FCE, SAST, DBNet, and DBNet++ had almost no misdetections, but all of them had quite a lot of missed detections. The TCM-DBNet had almost no missed detections, but quite a lot of misdetections, presumably because the introduction of the TCM made the model's learning capability. It is worth mentioning that the misdetections of the TCM-DBNet can be divided into two cases: the first is a small misdetection box, which covers less content; the

second is a large misdetection box, which contains some rough lines without much information. These two cases will not be recognized when the annotation recognition takes place, i.e., the recognition result is empty, which has little effect on the final extraction effect, but will increase the load and efficiency of the whole method. The model proposed in this paper had only a small number of missed cases, and the overall effect was better than the other comparison models, which is also basically consistent with the results in Table 1.

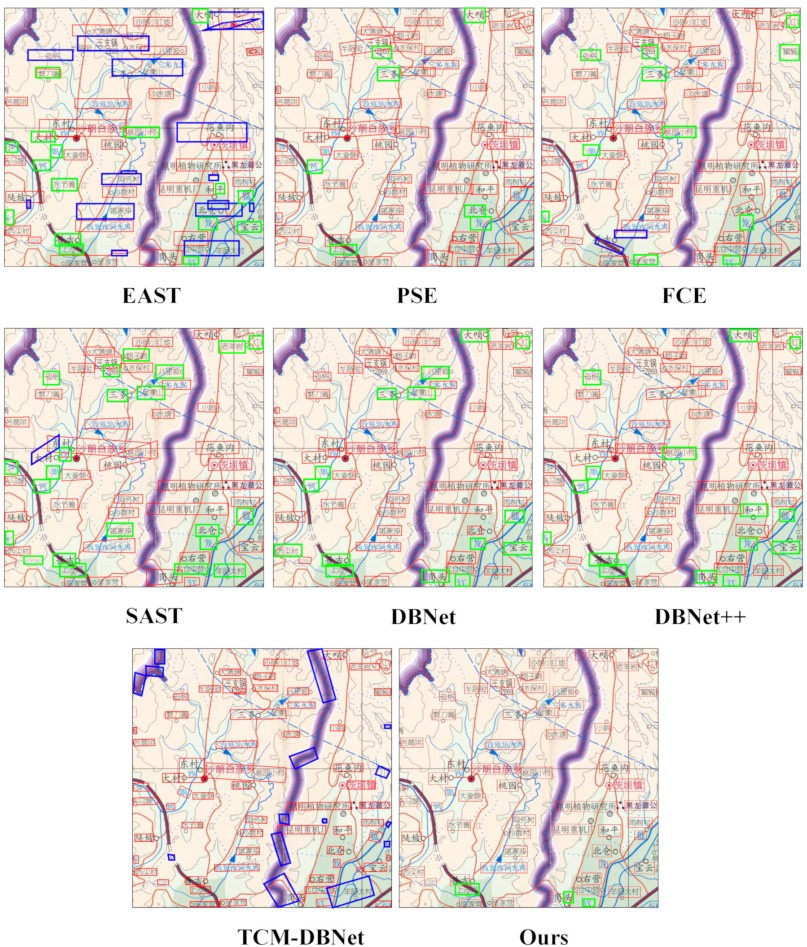

**Figure 13.** Comparison of detection results of different detection models. The red box indicates a correct detection, the blue box indicates an incorrect detection, and the green box indicates a missed detection.

The results of the annotated image recognition are shown in Figure 14. It is clear that the model used in this work exhibited comparatively accurate annotation picture recognition. The two most common categories of recognition errors were the following: (1) Interference from lines or other map features: The fifth annotation image in Figure 14 best illustrates this type of error. The recognition result contains an additional character or characters, because the detection area also included some other map elements. (2) Complex characters or similar-looking characters: For instance, in the sixth annotation image in Figure 11, the character that should have been recognized as '彝' was instead recognized as '舞'. These two characters are visually similar and belong to the category of similar-looking characters. In cases where the image resolution is not high, the model is more prone to making recognition errors.

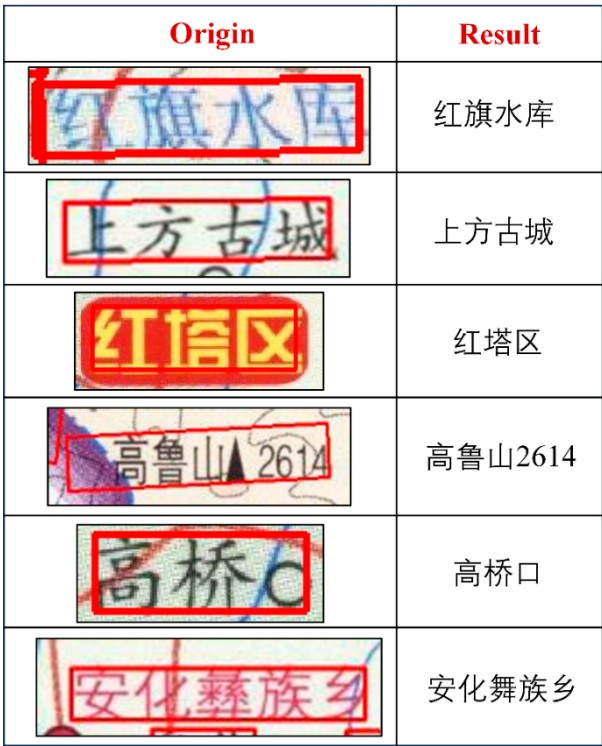

| Origin | Result |
|---|---|
| | 红旗水库 |
| | 上方古城 |
| | 红塔区 |
| | 高鲁山2614 |
| | 高桥口 |
| | 安化舞族乡 |

**Figure 14.** Chinese scanned map annotation recognition results.

## 5. Conclusions

This article proposed a deep-learning-based method for extracting Chinese scanned map annotations. The method mainly consisted of two modules: an improved EAST annotation detection module and a CRNN annotation recognition module based on transfer learning. The experimental results lead to the following conclusions:

(1) On the dataset of Chinese scanned maps, the suggested improved EAST annotation detection model in this research received ratings of 0.8990, 0.8389, and 0.8635 for precision, recall, and h mean, respectively. In terms of precision, recall, and h mean, these results are all more than 0.4 greater than those of the original EAST model. This proves that the model improvement suggested in this article was successful. Additionally, the model outperformed the control group in terms of the recall and h-mean metrics. The suggested model demonstrated gains for the corresponding metrics ranging from $-0.0354$ to 0.0907, 0.0131 to 0.2735, and 0.0467 to 0.1919 when compared with the models in the control group. This indicates the potential of the suggested model to deliver successful outcomes in the domain.

(1) The CRNN annotation recognition model based on transfer learning proposed in this article achieved scores of 0.9320, 0.8956, and 0.9134 for precision, recall, and h mean, respectively, on the Chinese scanned map dataset. In terms of the precision, recall, and h mean, these results are all more than 0.27 greater than those of the model without transfer learning. This shows the efficacy of the transfer learning strategy employed in the model suggested in this article. Additionally, the model outperformed the control group in all assessment measures (precision, recall, and h mean), with improvements for each metric ranging from 0.0294 to 0.1049, 0.0498 to 0.1975, and 0.0402 to 0.1582, respectively. This indicates once more how successfully the model suggested in this article can carry out annotation recognition tasks on Chinese scanned maps.

The extraction of Chinese scanned map annotations has been effectively accomplished in this paper, but there are still the following issues: Some annotations will be incorrectly detected during their recognition, even though these incorrectly detected parts may not al-

ways be recognized as characters. This will affect the final extraction results and still cause errors in the results. Future studies will take into account the following two main directions: how to further enhance note extraction performance and how to process spatial reconstruction and annotation extraction data to progress the process of map annotation vectorization.

**Author Contributions:** Conceptualization, Xun Rao and Jiasheng Wang; data curation, Xun Rao, Zhe Zhao and Jiasheng Wang; formal analysis, Xun Rao; investigation, Xun Rao, Zhe Zhao, Wenjing Ran and Mengzhu Sun; methodology, Xun Rao; project administration, Xun Rao and Jiasheng Wang; Software, Xun Rao; supervision, Jiasheng Wang; writing—original draft preparation, Xun Rao; writing—review and editing, Xun Rao and Jiasheng Wang; funding acquisition, Jiasheng Wang. All authors have read and agreed to the published version of the manuscript.

**Funding:** This research was funded by the National Natural Science Foundation of China, grant number 41961056.

**Data Availability Statement:** The code used in this study is available by contacting the corresponding author.

**Acknowledgments:** We would like to thank the anonymous reviewers for contributing to improve this manuscript, as well as the editors for their kind suggestions and professional support.

**Conflicts of Interest:** The authors declare no conflict of interest.

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
