# Peer review of "Deep-Learning-Based Annotation Extraction Method for Chinese Scanned Maps"

_ijgi, doi:10.3390/ijgi12100422_

Round 1
Reviewer 1 Report
This study proposes an automatic map annotation recognition method based on deep learning. The topic is both interesting and important, particularly for the analysis of ancient maps. The paper's structure is clear and easily comprehensible. However, there are a few concerns that the authors may consider:
The authors have developed a simulated map generator to construct a training dataset. Could you please clarify the necessity for simulation data? Is it due to the insufficiency of existing scanned map datasets? It would be helpful to provide more details about the simulation results. Additionally, it is important to ascertain whether the cartographic style of the simulated maps is consistent with that of real scanned maps. Requesting additional information on this matter.
It is recommended to replace the "Input feature map" and "Fused feature map" in Figure 5 with map-related images for better visual representation.
In Figure 10, there are instances of false and missing detections. It is suggested that the authors differentiate these from the correct detections, such as by using distinct colors or markers.
Figure 10 solely showcases the detection results of the proposed method. Would it be possible for the authors to include the detection results of other methods for a broader comparative analysis?
Regarding lines 315-317, it would be beneficial to explain why they are divided into three groups. What are the characteristics and differences among the data in each group? Furthermore, is it appropriate to employ simulated scanned maps for training and real scanned maps for testing purposes? Clarification on this matter is recommended.
Reviewer 2 Report
Strengths:
1. Clear Introduction: The article begins with a well-structured introduction that provides a solid understanding of the problem being addressed - the extraction of annotations from Chinese scanned maps. This helps readers comprehend the context of the research.
2. Dataset Construction: The article thoroughly explains the process of constructing both real scanned map and simulated map datasets, which is crucial for replicability and understanding the data used in the experiments.
3. Transfer Learning Explanation: The article offers a clear explanation of the transfer learning approach used in the research, making it accessible to readers with varying levels of expertise.
4. Experimental Results: The experimental results are presented in an organized manner with tables and figures. The use of multiple evaluation metrics (precision, recall, hmean) provides a comprehensive assessment of the proposed models.
5. Performance Improvements: The article highlights significant improvements in precision, recall, and hmean values achieved by the proposed models compared to existing models. This demonstrates the effectiveness of the research.
6. Discussion of Challenges: The article discusses the specific challenges related to Chinese scanned map annotation extraction, such as the complexity of Chinese characters and interference from map elements, providing valuable context for the research.
Suggestions for Corrections and Changes:
1. Clarity of Terminology: The article uses terms like "annotation detection" and "annotation recognition." While these terms make sense within the context of the article, it might be helpful to provide a brief explanation or definition of these terms for readers who are not familiar with them.
2. Data Source Citations: When discussing the source of the dataset, it would be beneficial to include proper citations or references to acknowledge the origin of the data. This enhances transparency and credibility.
3. More Detailed Model Architectures: While the article mentions the use of an "improved EAST model" and a "CRNN annotation recognition model," providing more detailed information about these models or references to their original sources would be beneficial for readers interested in replicating or extending the research.
4. Explain Unusual Results: In the discussion of the results, the article mentions "unusual results" for certain models like PSE. It would be helpful to briefly explain why these results are considered unusual to provide better insight into the evaluation.
Additions for Improvement of Methods and Results:
1. Comparison with State-of-the-Art: To demonstrate the research's advancement, consider comparing the proposed models with the current state-of-the-art models for similar tasks, especially if such models exist in the field of map annotation extraction.
2. Visualization: Including visual examples of successful and challenging cases of annotation extraction from Chinese scanned maps could enhance the reader's understanding of the method's performance.
3. Error Analysis: A more detailed error analysis, discussing common failure cases and potential solutions, would be valuable for future research and practical applications.
4. Scalability Discussion: Address how the proposed method scales when dealing with a larger number of annotations and larger maps. This could help readers understand the method's potential limitations.
5. Open Access to Code and Datasets: Consider sharing the code and datasets used in the research through open-access repositories or platforms, promoting transparency and enabling others to replicate and build upon the work.
Overall, the article shows promise in addressing the challenging task of Chinese scanned map annotation extraction, but it could benefit from improved clarity, detailed model explanations, and more in-depth analysis of results and challenges.
Reviewer 3 Report
Based on the deep learning method, this paper proposes a method to detect and recognize the Chinese map annotation. On the whole, the logic of the article is clear, and the methods and conclusions are reliable. Here are some specific suggestions:
(1) Will the proposed method be better for the recognition of English maps than Chinese maps? When simulating a map, the authors could also generate some English characters for such a comparison.
(2) In lines 169-171,denoising is not performed unless the noise is severe. How to judge whether the noise is severe?
(3) What are the improvements to EAST model in this paper?
(4) What is the training accuracy and testing accuracy of each model in this paper?
Round 2
Reviewer 1 Report
Dear authors, thank you for your improvements.